# Diagnosing Time-Varying Harmonics in Low-k Oxide Thin Film (SiOF) Deposition by Using HDP CVD

**DOI:** 10.3390/s23125563

**Published:** 2023-06-14

**Authors:** Yonggyun Park, Pengzhan Liu, Seunghwan Lee, Jinill Cho, Eric Joo, Hyeong-U Kim, Taesung Kim

**Affiliations:** 1School of Mechanical Engineering, Sungkyunkwan University, Suwon 16419, Republic of Korea; ygpark4392@skku.edu (Y.P.); pengzhan@skku.edu (P.L.); hwann96@g.skku.edu (S.L.); whwlsdlf94@skku.edu (J.C.); 2ComdelKorea, Ltd., 120 Heungdeokjungang-ro, Giheung-gu, Yongin 16950, Republic of Korea; jooninsu@gmail.com; 3Department of Plasma Engineering, Korea Institute of Machinery and Materials (KIMM), Daejeon 34103, Republic of Korea; 4SKKU Advanced Institute of Nanotechnology (SAINT), Sungkyunkwan University, Suwon 16419, Republic of Korea

**Keywords:** plasma diagnosis, harmonics, time-varying, nonlinear characteristic, dual-capacitor-sheath

## Abstract

This study identified time-varying harmonic characteristics in a high-density plasma (HDP) chemical vapor deposition (CVD) chamber by depositing low-k oxide (SiOF). The characteristics of harmonics are caused by the nonlinear Lorentz force and the nonlinear nature of the sheath. In this study, a noninvasive directional coupler was used to collect harmonic power in the forward and reverse directions, which were low frequency (LF) and high bias radio frequency (RF). The intensity of the 2nd and 3rd harmonics responded to the LF power, pressure, and gas flow rate introduced for plasma generation. Meanwhile, the intensity of the 6th harmonic responded to the oxygen fraction in the transition step. The intensity of the 7th (forward) and 10th (in reverse) harmonic of the bias RF power depended on the underlying layers (silicon rich oxide (SRO) and undoped silicate glass (USG)) and the deposition of the SiOF layer. In particular, the 10th (reverse) harmonic of the bias RF power was identified using electrodynamics in a double capacitor model of the plasma sheath and the deposited dielectric material. The plasma-induced electronic charging effect on the deposited film resulted in the time-varying characteristic of the 10th harmonic (in reverse) of the bias RF power. The wafer-to-wafer consistency and stability of the time-varying characteristic were investigated. The findings of this study can be applied to in situ diagnosis of SiOF thin film deposition and optimization of the deposition process.

## 1. Introduction

The scaling of semiconductors has led to the transformation of semiconductor devices, resulting in the development of technologies with shrinking device footprints, multilevel interconnecting stacks, and 3D architectures. However, this transformation has presented technical and productivity challenges in semiconductor manufacturing, such as gap-fill and leakage current [1]. To address these issues and improve productivity, the semiconductor industry has introduced inductively coupled plasma (ICP), which can generate high-density plasma (HDP) at low pressure (LP) [2,3]. Low-k dielectric materials have been applied into the back-end-of-line (BEOL) stage of semiconductor manufacturing to reduce parasitic capacitance. In addition, low-k dielectric materials have been used to reduce parasitic capacitance and improve signal propagation speed, and reduce signal delay, crosstalk between adjacent interconnects, and power consumption. Common low-k dielectric materials include organosilicate glass (OSG), carbon-doped oxide (CDO), and fluorine-doped silicon oxide (FSG), such as SiOF, which can be tailored to meet different manufacturing requirements.

ICP technology has become a widely used method for depositing dielectric insulators in various semiconductor manufacturing processes, such as shallow trench isolation (STI), pre-metal dielectric (PMD), interlayer dielectric (ILD), and intermetal dielectric (IMD) [2,4,5,6,7]. However, HDP intensifies plasma-induced damage (PID), which refers to the damage caused by electron charging in the device. This can decrease the device yield through photo conduction current activation by UV light and gate oxide degradation due to radio frequency (RF) coupling. To prevent these issues, UV-blocking layers and thin liners have been introduced [8,9,10,11,12,13,14,15,16]. In addition, the capacitance of the thin film can alter the electron charging amount, relaxation time, and electron density in the sheath [17,18,19].

ICP technology has several advantages, including high plasma density, uniformity, and productivity. It is introduced to the semiconductor fabrication processes by applying RF to coil-type antennas and bottom electrodes. One typical application of ICP technology is chemical vapor deposition (CVD) with HDP. Depending on the power frequency, the HDP CVD system has two methods of generating harmonics. The first origin of harmonics in HDP CVD is from the nonlinear Lorentz force that results from the interaction of the electric field and induced magnetic field in the coil. This force is magnified by low frequency (LF) rather than high frequency (HF) and has a greater effect on electron dynamics in the collision-less sheaths that form at LP. LF and LP generate and magnify the nonlinear effect in inductively coupled discharge. Since the charged particles, especially electrons, are subtly accelerated and decelerated by the electric and magnetic fields, the nonlinear characteristic has a profound effect on the electron dynamics [20,21,22,23,24].

The use of RF-driven electrodes in HDP CVD enables direct interaction between the bottom electrode and plasma through the sheath. When an RF electric field is applied, plasma oscillations with sheath expansion and collapse occur simultaneously. The time-varying behavior of the sheath determines the dynamics of the electrons approaching and occurring within it, which in turn primarily affects the harmonics [23,25,26,27]. Therefore, the second (s) source of harmonics in HDP CVD is the result of the nonlinear characteristic of the sheath. The sheath acts as a capacitor owing to its thickness and its potential for time-varying behavior [28,29,30]. Furthermore, the nonlinear behavior of electron dynamics results in electronic wave radiation and harmonics. The analysis of the time-varying behavior of harmonics can serve as an advanced method for judging the quality of deposition in advance without additional characterization equipment.

The plasma environment contributes to the variation in its parameters, such as capacitance, impedance, and potential, in the semiconductor fabrication process [31]. It affects the serial resonance of electrons in the bulk plasma and the sheath through electron dynamics, leading to the generation of the harmonics due to fast response and collective behavior of electrons. These harmonics are more easily diagnosed by the Fourier transforming the current waveform than by analyzing the voltage signal [19,25,30,32]. Therefore, for the plasma-related process, such as deposition or dry etching, changes in the plasma environment and surface reactions on the substrate can be identified by analyzing the harmonics [33,34,35]. Probe diagnosis is typically divided into two categories: invasive and noninvasive methods. Invasive methods involve physical contact with the plasma, such as with a Langmuir probe, which can limit its application [32,36,37,38]. Recently, noninvasive methods, such as directional coupler, have been studied and applied to investigate plasma processes [33,39,40,41]. These methods do not require physical contact with plasma and can provide information on harmonics without affecting plasma distribution.

Harmonics in HDP CVD oxide deposition are a result of the nonlinear Lorentz force and the nonlinear effect of the sheath on the RF-driven electrode, with the electron charge and oxide thickness affecting the electrode’s harmonics. Therefore, it is crucial to have satisfactory experimental conditions to observe the time-varying behavior of harmonics during in situ process step variation.

The goal of this study was to investigate the time-varying behavior of harmonics produced by the nonlinear characteristics of SiOF oxide deposition in the HDP CVD system and to understand the mechanism of harmonic generation. The signal was detected using a noninvasive directional coupler and transformed into the frequency domain using fast Fourier transform (FFT). The time-varying behavior of the harmonics was identified using data processing to interpret changes in the plasma environment, various parameters, and surface states. Finally, the diagnostic method for harmonics using a directional coupler was applied to assess process abnormalities and equipment operation status. The time-varying behavior of harmonics at a particular frequency is expected to serve as a criterion for in situ diagnosis of the thin film deposition processes, which could significantly reduce the need for ex situ detection, thus improving productivity and yield.

## 2. Materials and Methods

### 2.1. Experimental Setup

The experimental setup and conditions are presented in Figure 1. Figure 1 shows a schematic of the 200 mm HDP CVD system (Centura Ultima HDP CVD, Applied Material, Santa Clara, CA, USA). In this system, 2 MHz LF power is applied to the top and side coils, and 13.56 MHz power is applied to the bottom electrode. The HDP CVD system is biased by the RF applied to the bottom electrode. The escape (ESC) function is used to prevent the substrate temperature rise caused by ion bombardment. Here, however, the ESC function was off to prevent charging damage. A dual directional coupler (Omni directional coupler, ComdelKorea Ltd., Yongin, Republic of Korea) was installed on the power transmission line to diagnose the time-varying characteristics of harmonics in the plasma chamber environment. The directional coupler was a dual directional coupler that can detect both forward and reverse power. The sensitivity of the omnidirectional coupler was greater than 50 dB. The maximum adaptive frequency was up to the 10th harmonic, which was 10 times the reference frequency generated from the forward power and reverse power. The signal collected using the directional coupler was transferred to the computer for data processing. Then, it was converted to the frequency domain through FFT. The generated harmonics were too weak to visualize the inherent value. The data processing system (ParaDias system, ComdelKorea Ltd., Republic of Korea) displayed the detected harmonic power intensity in a power ratio format, and its definition is expressed in Equation (1):(1)$$N_{th}\;harmonic\;power\;\left(\%\right)=\frac{N^{th}frequency\;power}{1^{st}fundamental\;frequency\;power}\times 100.\;$$

### 2.2. Experimental Condition

The HDP CVD system is equipped with a unique heating sequence to heat the chamber walls and substrate. This sequence involved LF power and Ar, O_2_ gas. The SRO deposition, USG deposition, and low-k oxide (SiOF) deposition were then performed continuously as shown in Figure 2a. Table 1 shows the sequence of the deposition process, while Table 2 outlines the experimental process conditions. The conditions for forming the 600-Å-thick USG and the 4000-Å-thick SiOF were set at a substrate temperature of 400 °C and a 0.73 Pa (5.5 mTorr).

Figure 1 shows that the HDP CVD system is equipped with three RF power sources, and Table 2 lists the power sequence applied for each step. The gas flow conditions for each step in the process are shown in Table 3. Initially, Ar and O_2_ gas are introduced into the plasma chamber, and then LF power is supplied to ignite the plasma. The throttle valve is then opened to stabilize the pressure, and plasma heating is achieved by increasing the LF power. During the SiOF deposition step, the RF power is turned on. SiH_4_ and SiF_4_ gases are used this deposition. A transition step is then performed to prevent charging damage, and the process is terminated by turning off the LF power.

## 3. Results and Discussion

The process of depositing SiOF involves several steps, including pressure stabilization, plasma ignition, preheating, SRO and USG deposition, SiOF deposition, and a transition step. The behavior of the harmonics during SiOF deposition in Figure 3, where Figure 3a,b demonstrate the observed time-varying behavior of the LF power harmonics, while Figure 3c,d exhibit the same time-varying behavior of the RF power harmonics generated by the high-frequency RF power applied to the bottom electrode. Equation (1) expresses the harmonics as a power ratio, and then the 1st harmonic is not indicated in Figure 3. Since the power value of the fundamental frequency differs between forward and reverse power, the focus is on the change in trend of harmonics rather than on absolute comparison.

### 3.1. The Time-Varying Behavior of LF Power Harmonics

ICP technology has been widely adopted in semiconductor fabrication processes owing to its advantageous nonlinear effects. Specifically, the interaction of electric and magnetic fields results in a nonlinear Lorentz force, which affects the electron dynamics more significantly than the electric field force [3,5,20,21,22,42]. The nonlinear Lorentz force increased under low pressure and low frequency conditions of several MHz, where there are fewer collisions between electrons and neutral atoms, making it possible to transfer kinetic energy to the plasma bulk. Moreover, the plasma provides nonlocal plasma heating and high density while enhancing plasma uniformity through the pinch effect [20].

The hydrodynamic model is used to explain the nonlinear nature of ICP, which is caused by the nonlinear Lorentz force and the nonlinear convective part of the electron inertia force. These forces can be found in the electron momentum equation as follows [20,22,43]:(2)mdvdt+mv·∇v−eE−ev×B+mνv=0.
where *m* is the mass of electron, v is the rf drift velocity, E is the rf electric field, B is the rf magnetic field, ν is the collisional frequency of electron. During each step of deposition process, a low frequency of 2 MHz is applied to the top and side coils, as shown in Figure 2b. The preheating step, which utilizes the nonlinear characteristics of ICP, is commonly performed to prevent electron charging damage during SRO and USG deposition using LF power. Figure 3 demonstrates the harmonics generated using LF power, including the time-varying behavior of the forward and reverse harmonics of LF power. These harmonics were identified using the directional coupler mounted on the top coil. The measured LF power harmonics show the time-varying behavior of the 2nd, 3rd, and 6th harmonics.

To generate plasma in the preheating step, the introduction of gas and the application of LF power lead to a rapid transformation in the plasma environment. The potential rise caused by LF power, along with the low pressure formed by opening the throttle valve, amplifies the impact of the nonlinear Lorentz force. As a result, electrons in the plasma exhibit nonlinear characteristics, moving with fewer collisions and richer trajectories.

The nonlinear Lorentz force consists of a potential and a solenoid part, and the time-varying behavior of the 2nd harmonic is synchronized with the LF power [20,43,44,45,46]. When the LF power rises in the side coil, it ascends in the potential gradient, resulting in a higher intensity of the even harmonics. Additionally, the potential gradient affects the temporal and spatial electron dynamics, resulting in the collective behavior of the electrons affecting the electromagnetic field formed by the LF power of the top coil, which causes the rise in odd harmonics. It can be broadly summarized that the generated even harmonics cause the odd harmonics to ascend and descend [20,22,43,45].

In the preheating step, the forward and reverse power harmonics have the same tendency. On the other hand, in the SiOF deposition step using RF power, the time-varying behavior of harmonics changes slightly. The dynamics of electrons govern the time-varying behavior of harmonics in ICP techniques, where the nonlinear Lorentz force is the underlying mechanism. The plasma density and impedance affect the electron dynamics, such as the mean free path and the collision frequency of electrons.

The 6th harmonic is generated in the transition step, a discharging step in the plasma composed of Ar and O_2_. It is intended to reduce electron charging when RF power is off. Therefore, it shows the time-varying behavior of the 6th harmonic corresponding to the O_2_ plasma condition.

The time-varying behavior of LF power harmonics is plasma information that is synchronized to the variables, such as the power, pressure, and gas type, which affect the plasma. This change includes both the harmonic intensity and order. These data changes enable precise plasma diagnostics with data processing in the order of 0.1 s.

### 3.2. The Time-Varying Characteristics of RF-Driven Electrode Plasma Harmonics

In the SiOF deposition step, RF power is applied to the bottom electrode. Figure 3c,d show the time-varying behavior of the RF power harmonics generated in the SiOF deposition step. Unlike the harmonics of LF power, the harmonics of RF power show a wider range in the 2nd to the 10th harmonics. However, as shown in Figure 4, only the 7th and 10th harmonics show time-varying behavior as the oxide deposition progresses. The frequencies are 94.92 MHz and 135.6 MHz, respectively. The interaction between the plasma and the oxide layer surface alters as the SiOF layer thickness increases. These interaction changes can be identified by the time-varying behavior of harmonics.

The RF-driven electrode generates harmonics due to the nonlinear characteristics of the sheath, with the mechanism differing based on the nonlinear Lorentz force. Two primary factors can explain this phenomenon. First, the time-varying change in thickness of the sheath caused by plasma oscillations, and second, the time-varying change in RF potential [28,29,30]. Both factors determine the capacitance of the sheath and contribute to its nonlinear characteristic. Additionally, the sine wave of the RF field applied to the bottom electrode causes the sheath to expand and collapse. The periodic motion of the sheath creates a process in which electrons gain energy and are accelerated toward the plasma bulk to transfer energy [23].

As depicted in Figure 1, the electrode is connected in series with the RF generator, the matching network, and the transmission line. It is exposed to the plasma and interacts with it, forming a sheath. Due to the RF electric field, the sheath is self-biased by the charging phenomenon on the electrode surface by high-speed electrons. When the oxide layer is deposited, it becomes electrically connected in series with the sheath, forming a dual capacitor [47,48]. The oxide layer also affects the series resonance frequency [31]. Therefore, to understand the generation mechanism of harmonics from RF-driven electrodes, it is necessary to comprehend the series resonance, electron charging on the oxide surface, and capacitance characteristics of the sheath.

#### 3.2.1. The Relation between Series Resonance Frequency and Harmonics

In an RF power delivery system, the plasma reactor acts as a load that comprises both real and imaginary parts of impedance. The RF-driven electrode method involves applying RF to the bottom electrode, which directly interacts with the plasma and forms a sheath acting as a current rectifier and capacitor. As the oxide layer is deposited, the dual capacitance formed by the oxide layer and sheath changes with increasing thickness of the oxide [47,48], subsequently affecting the electrical characteristics of the power delivery and the series resonance. The relationship is expressed through the series resonance frequency $f_{resonance}$, as shown in Equation (3):(3)$$f_{resonance}=\frac{1}{2\pi \sqrt{LC}}.$$

The series resonance frequency of the power delivery system in the RF-driven electrode is affected by both inductance (*L*) and capacitance (*C*). The composite capacitance is formed by the sheath and deposited oxide layer and varies with the oxide film property and thickness on the wafer. The plasma system impedance has the lowest value and the highest current flow at the series resonance frequency [31]. Harmonics result from the collective behavior of electrons in the sheath and are represented as current flow in the series circuit. When the resonance frequency approaches the harmonics frequency, the harmonics characteristics can be derived from the plasma reactor impedance. The 7th harmonic frequency of 13.56 MHz is 94.92 MHz, which is close to the system resonance frequency of a 200 mm HDP CVD system [49]. Figure 4a shows the time-varying behavior of the 7th harmonic of the forward RF power. It demonstrates that, as RF power is applied, the plasma density increases, and the 7th harmonic closest to the system resonance frequency responds to the impedance. Employing a directional coupler, harmonics diagnosis can assess the series resonant frequency characteristics of the plasma reactor through a noninvasive method.

#### 3.2.2. The Time-Varying Behavior of Harmonics at the RF Driven Electrode

The HDP CVD system applies RF power to the bottom electrode where the wafer is placed to maximize the ion bombardment effect during the thin film deposition process. Physical etching occurs simultaneously due to ion bombardment; ion bombardment controls the incidence angle of the thickening materials that forms the overhang structure by ion sputtering. This results in excellent gap-fill properties with bottom-up growth, particularly in high aspect ratio structures [2,4,5,6,7,42]. As shown in Figure 2a, SRO and USG thin liners are deposited in advance to prevent PID. The USG liner must have conformal step coverage to prevent lateral current and possess sufficient insulation properties to prevent RF coupling by acting as an RF voltage divider. Consequently, controlling the thickness of the USG is crucial to ensure sufficient insulation properties. The time-varying behavior of the bias RF 10th (reverse) harmonic in Figure 4b can be explained by the double capacitor model and the particle conservation law. Figure 5a illustrates a schematic of a dual capacitor formed by the USG layer and sheath. The SiOF layer combines with the USG layer to form a composite capacitor. As the thickness of the SiOF increases, the capacitance of the composite dielectric exhibits time-varying behavior.

The capacitance of dual capacitor formed by the sheath and composite dielectric varies with the time following Equation (4):(4)$$C_{dual\;capaitor}\left(t\right)=\frac{C_{sheath}\times C_{total,\;dielectric}\left(t\right)}{C_{sheath}+C_{total,\;dielectric}\left(t\right)}$$

The total dielectric thickness increases with the deposition rate of SiOF on a 600-Å-thick *USG* liner using Equations (5) and (6):(5)$$d_{USG}=600,\;\left(t=0\right),$$
(6)$$d_{total,\;dielectric}\left(t\right)=d_{usg}\left(t=0\right)+depo.\;rate\times time\;\left(t:1\sim 77.5\;s\right).$$

The capacitance of the composite dielectric exhibits time-varying behavior according to film thickness, as described by Equations (7) and (8):(7)$$C_{dielectric}\left(t\right)=\varepsilon_{0}\varepsilon_{r}\frac{A}{d_{\;dielectric}\left(t\right)},$$
(8)$$C_{total,\;dielectric}\left(t\right)=\frac{C_{usg}\times C_{SiOF}\left(t\right)}{C_{usg}+C_{SiOF}\left(t\right)},$$$\mathrm{where}\;\varepsilon_{0}\;$is the vacuum permittivity, $\varepsilon_{r}\;$is the relative permittivity of dielectric, and *A* is the surface area.

Electron charging is an unavoidable phenomenon in oxide film deposition. The amount of charge *Q* on the oxide film is the product of dielectric capacitance and applied voltage. It follows the time-varying behavior of the composite capacitor, as shown in Equation (9).
(9)$$Q_{dielectric}\left(t\right)=C_{total,\;dielectric}\left(t\right)*V,$$
where *V* is the sheath potential, which is assumed here to be 400 volts.

Figure 5b depicts the relationship between capacitance and electron charge (*Q*) as a function of increasing SiOF thickness. The low-k oxide is deposited to a thickness of 4000 Å on a 600 Å-thick USG, with a dielectric constant of 4.0, while the SiOF has a dielectric constant of 3.5 [50,51]. The substrate area is assumed to be the same as that of a 200 mm wafer. As the SiOF thickness increases, the capacitance and *Q* of the composite oxide exponentially decrease.

The absence of collision between the electrons and neutral atoms in the sheath, along with the electron mean free path being larger than the reactor diameter, occurs when the pressure is below 10 Pa. The expansion and collapse of the sheath, driven by the RF field, significantly affects the acceleration and deceleration of electrons [23]. Surface charging of electrons can lead to a change in the electron density within the sheath, which in turn affects the intensity of harmonics.

The electrons in the sheath become charged when they are on the surface of the oxide film. However, when the sheath expands, the electrons gain energy and accelerate into the plasma bulk. Assuming that the process parameters remain constant, the plasma bulk density and the electron density flowing into the sheath remain constant. The variation in electron density due to sheath oscillation can be explained by the current conservation law, also known as particle conservation [23].

Figure 6 illustrates the behavior of electrons in the plasma bulk and the sheath. The sheath serves as a medium for electron movement on the surface. The flow of electrons into the sheath can be represented as the sum of the electrons charged on the surface and the electrons returning to the plasma bulk, as shown in Equation (10). However, it is assumed that no secondary electrons are generated from collisions caused by the acceleration of electrons and ions.
(10)$$n_{e,i}=n_{e,Q}+n_{e.r}$$

Here, $n_{e,i}$ is the electron density flowing into the sheath, $n_{e,Q}$ is the electron density charged by the oxide capacitance, and $n_{e.r}$ is the electron density returned to the plasma bulk by sheath oscillation.

The amount of surface charging, Q, shown in Figure 5b, decreases exponentially as the SiOF thickness increases. Meanwhile, the 10th harmonic bias RF at 135.6 MHz increases exponentially in Figure 4b. To elucidate the electron dynamics within the dual capacitor created by the sheath and oxide layer, electron mobility must be considered. Electrons within the solid state of the oxide layer are trapped by surface charging, where in the free space sheath, the electron mobility is high enough to allow for acceleration to the speed of light without colliding. Based on the particle conservation relation in Equation (10), the reduction in surface charging ($n_{e,Q}$) leads to an increase in electron density ($n_{e.r}$) returning from the sheath to the plasma bulk. Therefore, the time-varying behavior of the 10th harmonic in Figure 3b can be interpreted as a process involving electron acceleration to the electrode surface, a decrease in electron density due to surface charging [17,18], and an increase in the density of the returning electrons. The 10th harmonic reflects the dynamics of free electrons in the sheath, which is free space. Thus, the change in harmonic intensity is related to the change in the electron density in the sheath after interacting with the surface.

The time-varying behavior of the 10th harmonic can be used as an indirect indicator of USG thickness, as it is affected by the dynamics of electrons in the sheath and their interaction with the surface. Although USG thickness cannot be precisely controlled due to the plasma heating effect, monitoring the behavior of the 10th harmonic can provide a means to indirectly manage USG thickness and enhance process yield.

The study of electron dynamics in relation to electromagnetic waves has long been a topic of significant research interest. The Larmor formula defines the relationship between electrons and electromagnetic waves [52,53]. A collision-less sheath formed at low pressure allows electrons to accelerate to light speed without experiencing collisions. In the same condition, the electromagnetic wave is generated when charged particles are accelerated by an external field. The power equation can be expressed as:(11)$$P=\frac{2e^{2}}{3}\frac{a^{2}}{C^{3}}$$
where *e* is the electron charge, *a* is the acceleration speed, and *c* is the speed of light. The electromagnetic wave power is proportional to the square of the electric charge and the square of the acceleration. The electromagnetic wave power is generated by the collective behavior of the electrons in the sheath. The time-varying behavior of the 10th (reverse) RF harmonic is effectively represented by the dual capacitor model and particle conservation model investigated in this study.

## 4. Conclusions

In conclusion, the time-varying behavior of harmonics during SiOF deposition using an HDP CVD system was analyzed using a dual directional coupler and data processing system. The assessment was carried out continuously throughout the process and was found to be synchronized with changing process conditions. Specifically, the increase in LF power and pressure change was found to be associated with an increase in the 2nd and 3rd harmonics, which were caused by the nonlinear Lorentz force resulting from the interaction of the electric and magnetic fields and the nonlinear electron convection. Additionally, the 7th and 10th harmonics of RF power were detected, and their mechanisms elucidated, with the 7th harmonic associated with the series resonance resulting from the time-varying behavior of the dual capacitor composed of the plasma sheath and oxide film, and the 10th harmonic caused by the time-varying electron charging on the oxide layer surface and the electron dynamics in the sheath. The behavior of harmonics was found to be an indicator of plasma parameters and electron dynamics, which can be utilized as wafer process information and potentially be employed as a criterion for determining the start of subsequent processes.

## Figures and Tables

**Figure 1 sensors-23-05563-f001:**
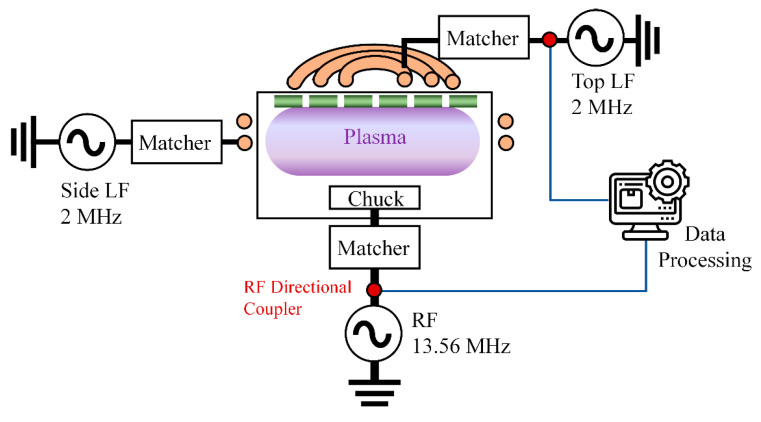
Schematic diagram of HDP system with harmonic diagnosis.

**Figure 2 sensors-23-05563-f002:**
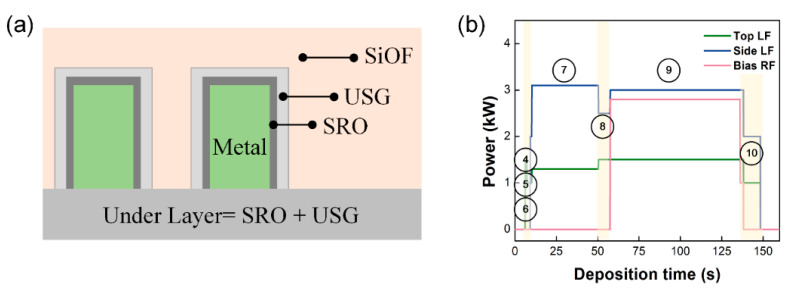
(**a**) The schematic structure of ILD. (**b**) The sequence of LF and RF power.

**Figure 3 sensors-23-05563-f003:**
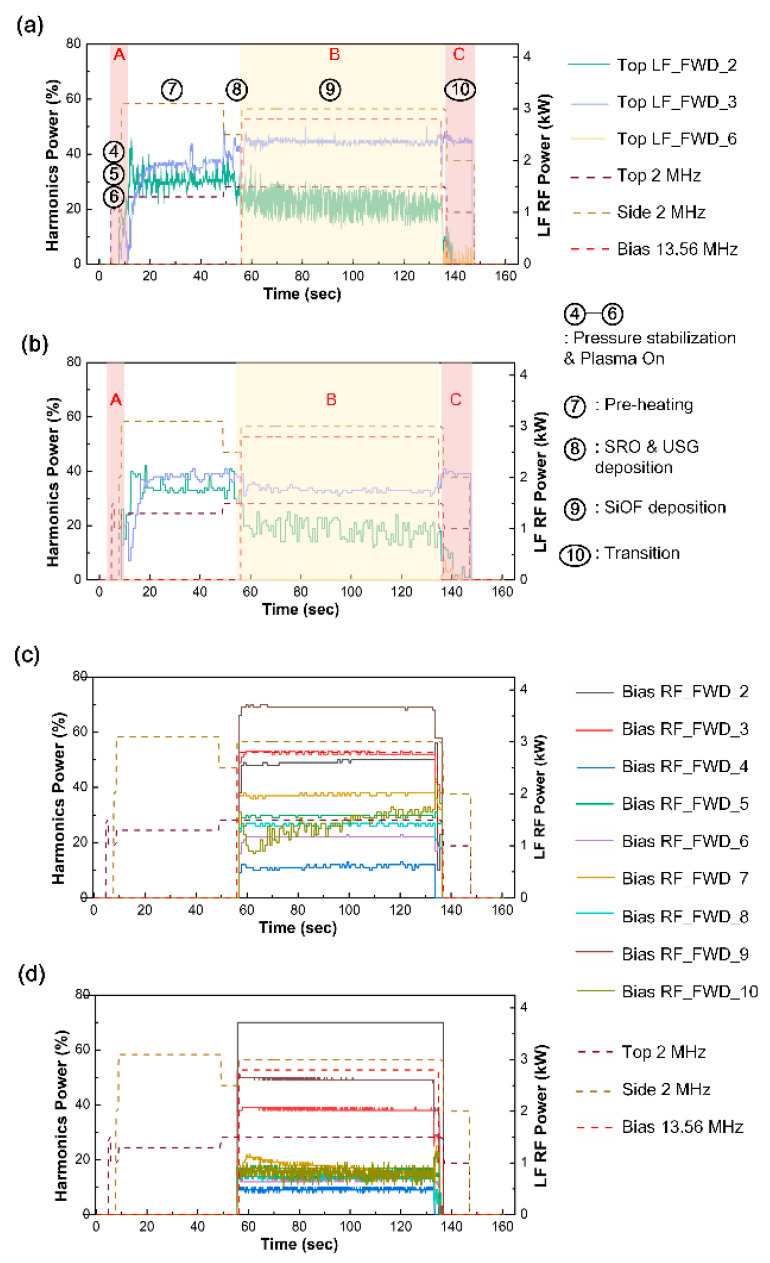
The time-varying behavior of (**a**) forward harmonics of LF power, (**b**) reverse harmonics of LF power, (**c**) forward harmonics of bias RF power, and (**d**) reverse harmonics of bias RF power.

**Figure 4 sensors-23-05563-f004:**
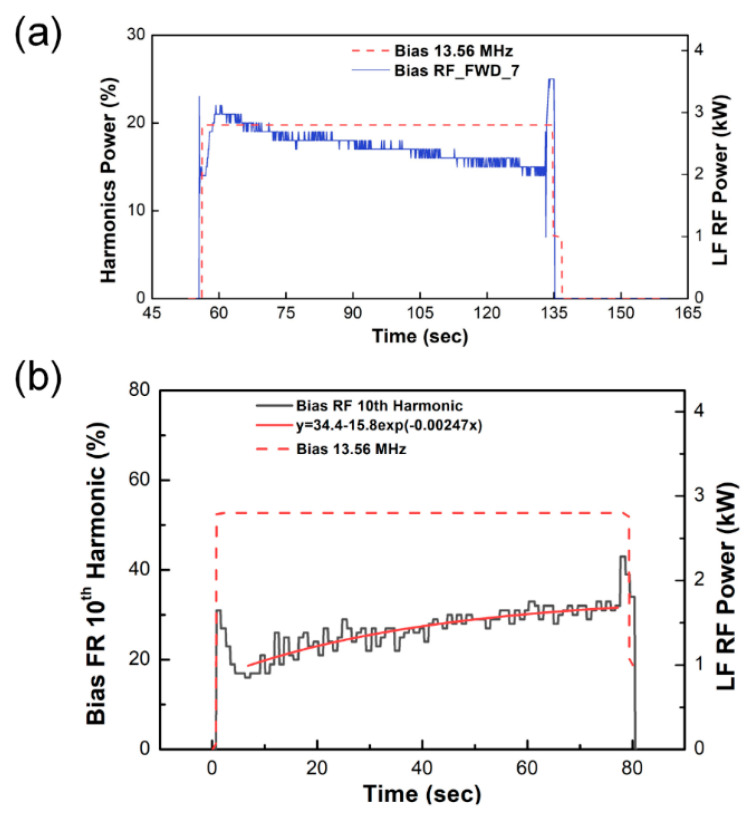
The time-varying behavior of (**a**) 7th bias RF harmonics (forward) (**b**) 10th bias RF harmonics (reverse).

**Figure 5 sensors-23-05563-f005:**
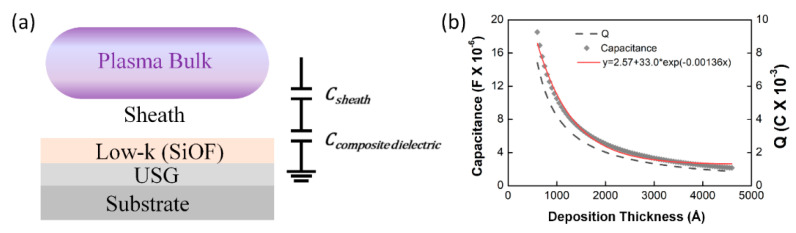
(**a**) The schematic illustration of a dual capacitor structure. (**b**) The relationship between total capacitance and electron charging by dielectric thickness.

**Figure 6 sensors-23-05563-f006:**
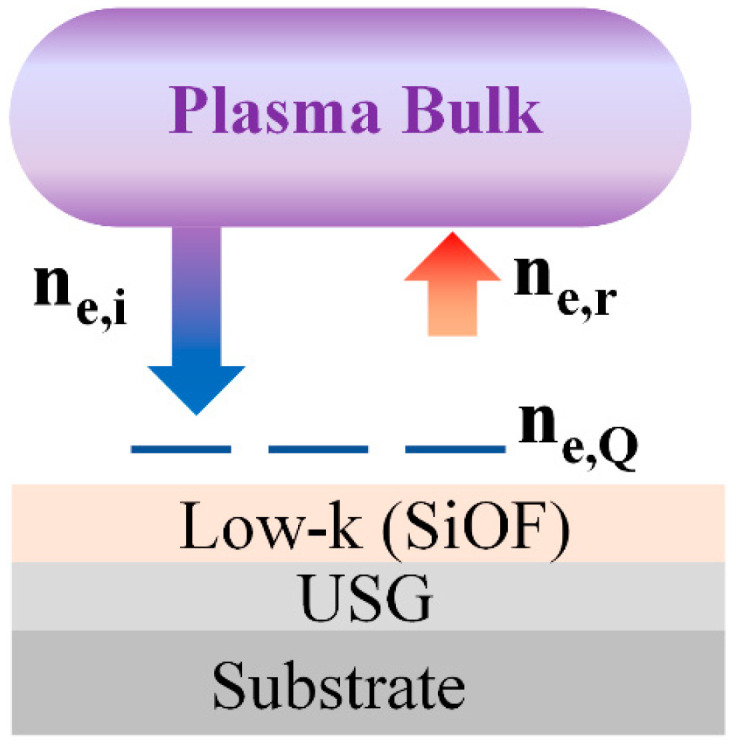
The behavior of electrons in the plasma sheath.

**Table 1 sensors-23-05563-t001:** The sequence of low-k oxide (SiOF) deposition.

No.	Step	Gas	Power
1	Chamber cleaning	SF_6_	Microwave (GHz)
2	Chamber seasoning	SiH_4_, O_2_, Ar	Top and side LF
3	Wafer loading	---	---
4	Pressure stabilization	Ar, O_2_	---
5	Plasma on	Ar, O_2_	Top and side LF
6	Throttle valve on	Ar, O_2_	Top LF ramp-up
7	Preheating for heat-up	Ar, O_2_	Top and side LF
8	SRO and USG deposition	SiH_4_, Ar, O_2_	Top and side LF
9	SiOF deposition	SiH_4_, SiF_4_, Ar, O_2_	Top and side LF, RF
10	Transition	---	Reduced top and side LF, RF
11	Power and gas off	---	---

**Table 2 sensors-23-05563-t002:** SiOF deposition condition with 600 Å thickness of USG.

Step	Time (s)	Thickness (Å)	Power	Condition (kW)
Heat-up	40.9	--	Top, side, bias (RF)	1.3, 3.1, 0
SRO deposition	2	600	Top, side, bias (RF)	1.5, 2.5, 0
USG deposition	5.1		Top, side, bias (RF)	1.5, 2.5, 0
SiOF deposition	77.5	4000	Top, side, bias (RF)	1.5, 3, 2.84

**Table 3 sensors-23-05563-t003:** Gas flow rate of SiOF deposition process.

No.	Step	Ar	Ar (Top)	O_2_	SiH_4_ (Side)	SiH_4_ (Top)	SiF_4_
1	Gas turn on	110	16	--	--	--	--
2	Side RF power on	110	16	--	--	--	--
3	Open throttle VV	110	16	--	--	--	--
4	Heat up (preheat)	110	16	110	--	--	--
5	SRO deposition	110	16	110	--	20	--
6	USG deposition	60	5	120	42	4	--
7	SiOF deposition	60	5	120	34	3.5	40
8	SiH_4_ gas off	60	5	120	--	--	--
9	Transition	60	5	120	--	--	--
10	De-chuck	110	16	120	--	--	--

## Data Availability

Not applicable.

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
