# Peer review of "Diagnosing Time-Varying Harmonics in Low-k Oxide Thin Film (SiOF) Deposition by Using HDP CVD"

_sensors, 2023, doi:10.3390/s23125563_

Round 1

Reviewer 1 Report

This paper analyzes the harmonic characteristics in a high-density plasma chemical vapor deposition for the deposition of low-k SiOF. The harmonics are influenced by LF power, pressure, gas flow rate and underlying layer. The findings presented in this paper can be applied to diagnose and optimize SiOF deposition process. The experimental results were interesting and were well-presented, so I would like to suggest accept this paper after solving some minor concerns as below (minor revision recommended):

1. How is the behavior of the 10th harmonics of USG layer changed with different thickness?

2. Also, is the behavior of the 10th harmonics reproducible?

3. How is the diagnostic process of an RF module using harmonics conducted?

4. In conclusion part, please give a more details for "as a criterion for the start of subsequent processes".

5. (1) Please change "schematic" into "schematic diagram" in Figure 1 caption.

(2) Please change "8. SRO & USG" into "8. SRO & USG deposition" in Figure 3(b).

(3) Please change "schematic" into "schematic illustration" or "schematic structure" in Figure 5(a) caption.

Author Response

  1. Response to Comments of Reviewer #1

This paper analyzes the harmonic characteristics in a high-density plasma chemical vapor deposition for the deposition of low-k SiOF. The harmonics are influenced by LF power, pressure, gas flow rate and underlying layer. The findings presented in this paper can be applied to diagnose and optimize SiOF deposition process. The experimental results were interesting and were well-presented, so I would like to suggest accepting this paper after solving some minor concerns as below (minor revision recommended).

Q1. How is the behavior of the 10th harmonics of USG layer changed with different thickness?

A1) We thank the reviewer’s critical comments. When the USG thickness is reduced to 300 Å, the behavior of the 10th harmonic does not exhibit an initial decrease followed by an exponentially increasing trend over time. Instead, it shows an initial increase to a certain level of intensity, which remains relatively constant as time progresses. In the case of a thin thickness, the USG oxide exhibits characteristics similar to metals due to its inadequate insulation resistance.

Figure R1. Comparative evaluation data on the behavior of the 10th harmonic

Q2. Also, is the behavior of the 10th harmonics reproducible?

A2) We thank the reviewer’s helpful comments. The behavior of the 10th harmonic in the 1X process (1 deposition and cleaning step) demonstrates reproducibility.

Figure R2. Representative of behavior of 10th harmonic for reproducibility

Q3. How is the diagnostic process of an RF module using harmonics conducted?

A3) We thank the reviewer’s critical comments. The diagnosis of the RF module is contingent upon the correlation between the electron density of the plasma, formed by the output power of the RF generator entering the chamber, and the matching efficiency of the matcher. A decrease in RF power results in a reduction of electron density and subsequently leads to a diminished intensity of generated harmonics.

Q4. In conclusion part, please give a more details for "as a criterion for the start of subsequent processes".

A4) We thank the reviewer’s critical comments. The thickness of the USG layer varies depending on the seasoning, and when the 1X process is carried out continuously, the temperature of the chamber wall increases, leading to changes in the thickness of the USG layer and alterations in the behavior of the 10th harmonic. By monitoring the changes in the behavior of the 10th harmonic, it is possible to perform Virtual Metrology (VM), enabling the determination of whether subsequent processing should proceed.

Q5.

(1) Please change "schematic" into "schematic diagram" in Figure 1 caption.

(2) Please change "8. SRO & USG" into "8. SRO & USG deposition" in Figure 3(b).

(3) Please change "schematic" into "schematic illustration" or "schematic structure" in Figure 5(a) caption.

A5-a) We thank the reviewer’s helpful comments. As you mentioned, we modified in the revised manuscript.

A5-b) P.3, Line 136: Figure 1. Schematic diagram of HDP system with harmonics diagnosis.

A5-c) P.6, Figure 3: figure 3 was modified in revised manuscript.

A5-c) P.9, Line 303: Figure 5. (a) The schematic illustration of a dual capacitor structure (b) The relationship of total capacitance and electron charging by dielectric thickness.

Reviewer 2 Report

In their work, Park et al have reported on the diagnosis of time-varying harmonics in low-K SiOF films using HDP CVD process. While the article is interesting, there are fundamental challenges in the approach. 

1. One of the biggest problems of using the FFT to process the data is that it leads to an averaged spectrum which has been integrated over the whole acquisition time, invariably resulting in the loss of any transient information which they are hoping to extract. For analysis of dynamic time-varying processes I would rather recommend using Wavelet Transforms which can localise both the temporal as well as frequency-based analysis and do not rely on the averaging nature of the FFT. 

2. Considering the number of process variables, multi-variate analysis such as Principal Component Analysis (PCA) could provide a better approach to understanding the process much better. For instance,  the authors mention that "The time-varying behavior of LF power harmonics is plasma information that is synchronized to the variables, such as the power, pressure, and gas type, which affect the plasma." Whilst this may indeed be true, it does not tell the authors or indeed the readers which of the components actually affects the plasma or the deposition process. 

3. Does the temporal limit of 0.1 s arise from the data acquisition or from the data processing (considering that all the data is collected off-line and then analysed rather than being an on-line real-time processing)

Author Response

  1. Response to Comments of Reviewer #2

In their work, Park et al have reported on the diagnosis of time-varying harmonics in low-K SiOF films using HDP CVD process. While the article is interesting, there are fundamental challenges in the approach. 

Q1. One of the biggest problems of using the FFT to process the data is that it leads to an averaged spectrum which has been integrated over the whole acquisition time, invariably resulting in the loss of any transient information which they are hoping to extract. For analysis of dynamic time-varying processes I would rather recommend using Wavelet Transforms which can localize both the temporal as well as frequency-based analysis and do not rely on the averaging nature of the FFT. 

A1) The authors sincerely appreciate the invaluable comment. Wavelet transform, as mentioned, is a useful technique for temporal and frequency analysis. However, when applying wavelet transform, there are certain drawbacks to consider:

  1) The choice of wavelet used for the transformation encompasses a variety of options, and the selection of a specific wavelet can lead to different results. This introduces a dependency on the developer, resulting in different harmonic characteristics for the same phenomenon. This can lead to distorted analysis of harmonics or reliability issues in wavelet transform application.

  2) Additionally, the selection of wavelets should consider frequency characteristics. However, the changes in plasma characteristics within the process chamber and variations in the sheath formed at the surface are influenced by various factors. This makes it challenging to determine the frequency characteristics resulting from such changes in a deterministic manner.

  3) In process or process chamber diagnostics, real-time harmonic analysis needs to be conducted within a very short time frame using unstructured arbitrary input signals. This requires a different approach compared to post-analysis of signals.

On the other hand, the FFT method allows the decomposition of arbitrary input signals, acquired through a directional coupler, into a sum of periodic functions with various frequencies, using cosine and sine basis functions. This enables the examination of signal characteristics, including the diverse frequency and intensity variations caused by changes in plasma and sheath properties, in real-time. Although FFT has temporal limitations, in semiconductor process and chamber diagnostics, the temporal constraint can be complemented by utilizing the harmonic trend generated every 0.1 seconds through data processing.

Q2. Considering the number of process variables, multi-variate analysis such as Principal Component Analysis (PCA) could provide a better approach to understanding the process much better. For instance, the authors mention that "The time-varying behavior of LF power harmonics is plasma information that is synchronized to the variables, such as the power, pressure, and gas type, which affect the plasma." Whilst this may indeed be true, it does not tell the authors or indeed the readers which of the components actually affects the plasma or the deposition process. 

A2) The authors sincerely appreciate the invaluable comment. As mentioned, multivariate analysis such as Principal Component Analysis (PCA) can be employed to assess the influence of process parameters. Furthermore, understanding the trends of harmonics influenced by process parameters can facilitate easier process diagnostics. In this study, the harmonic diagnostics were conducted to investigate the synchronization with various potential variations that may occur during the process, aiming to explore real-time diagnostics for process processes and chamber diagnostics. The response regarding the synchronization between process parameters and LF power harmonics is as follows:

In the ICP equipment, LF power is applied to the top and side coils, creating an electrical and magnetic field that spatially forms. The gradient of the electrical and magnetic fields becomes the driving force for nonlinear motions, such as the ponderomotive motion of electrons. Pressure influences the ionization electron density, leading to collective behavior of electrons undergoing nonlinear motion, resulting in the generation of harmonics. These results demonstrate the changes in 2nd and 3rd harmonics due to variations in LF power and pressure. Ultimately, the variations in input parameters affect the behavior of harmonics (harmonic order, intensity), enabling their utilization as a diagnostic tool for plasma diagnostics and the TTTM (Tool to Tool Matching) of process chambers. As mentioned in the introduction, with the application of these characteristics, it is possible to diagnose anomalies in power delivery of the RF module, detect leaks using gas's inherent harmonics, and diagnose arcs in high-voltage and high-current systems.

Q3. Does the temporal limit of 0.1 s arise from the data acquisition or from the data processing (considering that all the data is collected off-line and then analyzed rather than being an on-line real-time processing)

A3) The authors sincerely appreciate the invaluable comment. In the context of the data collected through a directional coupler, which is obtained in analog format, it can be considered that there is minimal time involved in the data acquisition process. Considering that the data is collected offline and subsequently analyzed, the time required for data processing can be even lower than the temporal limit of 0.1 seconds. In this study, the temporal limit of 0.1 seconds is a value that takes into account the realistic and applicable fastest data communication speed and data storage capacity of the FDC (Fault Detection & Classification) system, when integrated for diagnostics in a semiconductor production line. The data processing tool utilized in this experiment allows for various settings ranging from 0.1 to 1 second.

Round 2

Reviewer 2 Report

I appreciate the author's response on the choice of FFT over WT considering the need for choosing the most appropriate mother/daughter wavelets, especially for real-time analysis. Having said that the FFT analysis shown in the manuscript too is being done on offline data and not real-time data!

I would recommend that the authors add the details of the FFT parameters including the sampling frequency etc. and any filtering that was undertaken on the data to the revised manuscript. 

Author Response

Authors’ Reply to the Reviewers' Comments on:

Manuscript ID: Sensors-2419799

Diagnosing Time-varying Harmonics in Low-k Oxide Thin-Film (SiOF) Deposition by using HDP CVD

By Yonggyun Park, Pengzhan Liu, Seunghwan Lee, Jinill Cho, Eric Joo, Hyeong-U Kim* and Taesung Kim*

We deeply appreciate the effort that the reviewers have taken in reviewing our manuscript. Changes have been carried out according to the comments and highlighted in Yellow in this revised version of the manuscript. We hope that our revisions properly addressed all the points of the comments.

  1. Response to Comments of Reviewer #2

Q1) I appreciate the author's response on the choice of FFT over WT considering the need for choosing the most appropriate mother/daughter wavelets, especially for real-time analysis. Having said that the FFT analysis shown in the manuscript too is being done on offline data and not real-time data!

A1) We thank the reviewer’s critical comments. I sincerely apologize for any misunderstanding caused. I would like to clarify that the data processing in this manuscript was conducted in real-time, rather than offline. The data processing system is designed to extract harmonics from arbitrary signals inputted from sensors at a 0.1 second (s) interval and display them on a Graphic User Interface (GUI). Therefore, the diagnostic graphs of harmonics were generated by graphing the harmonics data, which had been stored at a 0.1 s interval.

Q2) I would recommend that the authors add the details of the FFT parameters including the sampling frequency etc. and any filtering that was undertaken on the data to the revised manuscript. 

A2) We thank the reviewer’s critical comments. I sincerely apologize for the difficulty in accepting the suggestions proposed by the reviewer. I would like to express my apologies. The sampling frequency and parameters associated with the FFT transformation applied in this experiment have been optimized using extensive expertise to ensure high-speed transformation. As an author affiliated with a company, the focus of this paper was to present results that have undergone internal censorship and can be publicly disclosed. It was believed that the existing results were sufficient for adequate explanation. It is important to note that certain aspects of these expertise are difficult to be guaranteed as rights and therefore cannot be openly disclosed in this paper. Your understanding in this matter is greatly appreciated.
